# Study on Urban Thermal Environment in Beijing Based on Local Climate Zone Method

**Fei Han [1], Xinqi Zheng [1,*], Jiayang Li [1], Yuwei Zhao [1] and Minrui Zheng [2]**

1   School of Information Engineering, China University of Geosciences, Beijing 100083, China; 2004200030@cugb.edu.cn (F.H.); lijiayang@cugb.edu.cn (J.L.); 2104200074@cugb.edu.cn (Y.Z.)
2   School of Public Administration and Policy, Renmin University of China, Beijing 100872, China; minruizheng@ruc.edu.cn
*   Correspondence: zhengxq@cugb.edu.cn

**Abstract:** In recent years, with the introduction of the concept of a local climate zone (LCZ), researchers have proved that adding an LCZ to the Weather Research and Forecasting (WRF) Model can improve the simulation effect. However, many existing studies cannot explain whether the improvement of accuracy in the model results is the effect of the refined zone or the effect of urban area correction, so they cannot explain the advantages of LCZ data. Therefore, this paper uses remote sensing images to generate two kinds of land use data sets and introduces them into the Weather Research and Forecasting Model coupled with the building energy model (WRF-BEM). In this paper, the two factors of urban area expansion and fine classification are considered, and three numerical examples are set up to simulate high-temperature weather in August 2019. The research shows that the simulated 2 m temperature of the scheme of correcting only urban area is the closest to the observed data. Although the RMSE in the 2 m temperature simulated by the LCZ scheme is 0.43 °C higher than that of the scheme of correcting only the urban area, it can well reproduce the spatial variation characteristics of 2 m temperature. In addition, different urban morphologies affect the spatial distribution of the surface urban heat islands in Beijing. High surface urban heat island effect zones mainly appear in the compact low-rise, compact mid-rise, and large low-rise types.

**Keywords:** local climate zone; WRF model; urban canopy model; Beijing





## 1. Introduction

In recent years, China's urbanization process has gradually accelerated. By 2019, the urbanization rate has reached 60.6%. It is estimated that by 2025, the urbanization rate will reach 65.5%. The improvement of the urbanization level leads to the change of land cover and urban morphology, which brings a series of climate and environmental effects, such as the urban heat island effect, air pollution, and extreme weather process [1], which shows that the urban wind and heat environment poses a serious threat to residents' lives and health. The change of land cover is predominantly reflected in urban expansion, which changes the surface morphology. The urban surface morphology affects the surface energy balance, the exchange process of power, heat, and turbulence between the surface and the atmosphere, and then affects the climate environment [2,3]. The urban morphological structure includes the urban layout, materials, number and height of buildings, etc., [4]. With the acceleration of urbanization, the change in urban morphology and structure, such as the increase of building height, will reduce the urban canopy ventilation, leading to the aggravation of the urban thermal environment and environmental deterioration [5,6].

The urban thermal environment is mainly caused by the drastic evolution of urban underlying surface and human activities, which leads to the high concentration of urban heat [7]. In order to alleviate the negative impact of the urban high-temperature heat wave on residents, many scholars have explored the influence of the impervious layer, green space, and other spatial forms on the thermal environment from the physical characteristics

of the urban underlying surface [8,9]. The Weather Research and Forecasting (WRF) Model is widely used to simulate the urban thermal environment because it can consider land cover types, the urban spatial morphology and structure, etc. At present, the static data of the underlying surface used in WRF are divided into USGS (United States Geological Survey) data and MODIS (Modern Resolution Imaging Spectroradiometer) data [10]. These two data describe the land use situation of urban areas in China, which is not precise enough, lacks timeliness, and cannot truly reflect the urban underlying surface [11,12]. Therefore, more precise underlying surface data are needed to improve the simulation effect of the model. The local climate zone (LCZ) proposed by Stewart and Oke (2012) can classify the complex urban underlying surface in a refined way and apply the refined classification results to the model simulation to improve the forecast accuracy, which provides a new perspective for the study of the urban thermal environment. Their cities are divided into 17 categories according to the surface types, of which 10 are architectural categories and 7 are natural surface categories [13]. Some progress has also been made in the simulation of the urban thermal environment and the atmospheric environment by combining the LCZ map as the underlying surface and WRF model. It has been proven that adding LCZ to WRF can improve the simulation effect. Brousse et al. (2016) constructed the LCZ map of Madrid, Spain, and used the WRF-BEM model to simulate the local meteorological field. The results show that LCZ classification can improve the simulation performance of the model [14]; Mu et al. took Beijing as an example to discuss the influence of the LCZ map and model default underlying surface on the simulation ability. The research proved that the LCZ map can improve the simulation effect of the model on 2 m temperature [15]. The application of LCZ classification to the WRF model has been applied all over the world [16–19].

The WRF model provides an urban canopy model (UCM) to express the characteristics of urban morphology and structure. The urban canopy model can be divided into the single-layer urban canopy model (SLUCM), the building environment parameter (BEP), and the building energy model (BEM) [20–22]. They can well simulate the thermal and dynamic processes of urban boundary layers and meet the research needs of different disciplines [23]. The SLUCM scheme considers that all buildings have the same morphological characteristics, calculates the sensible heat flux of each surface, and considers the influence of human activities. The BEP scheme considers the morphological characteristics of different buildings and the process of energy exchange between buildings and the boundary layer. The BEM scheme increases the function of an air conditioning system based on BEP, which is used to consider the influence of air conditioning on the energy inside and outside the building. There are many current studies on the urban canopy model. It is found that the urban canopy model has a great impact on the simulation results of the urban climate and atmospheric environment [24,25].

The characteristic of urban areas is that the temperature rises relative to the surrounding undeveloped rural areas. This phenomenon is called the urban heat island effect. Meteorological observations in many cities around the world have quantified the heat island effect. Recent examples include studies by Bueno, Roth et al. (2014), Coch Roura et al. (2017), and Detommaso, Costanzo et al. (2021) [26–28]. In recent years, the research on mitigation measures of the urban heat island effect has been widely concerned [29]. As the urbanization process has changed the surface properties, building materials with low albedo and high specific heat capacity absorb a large amount of solar and infrared radiation, and the accumulated heat is released into the atmosphere, further increasing the temperature. Therefore, UHI is greatly affected by the thermodynamic properties of building materials, and changing the properties of the building envelope is a potential method to alleviate the impact of UHI [30]. The roof of a building accounts for about 20–25% of the urban surface, and the solar radiation received by the roof can easily heat the roof surface to 50–60 °C [31], which increases the risk of indoor overheating [32]. Therefore, the cooling roof has become an urban cooling technology concern in the current scientific research field [33]. This technology is mainly divided into two types: (1) increasing

the albedo of the roof and reducing the solar short-wave radiation received by the roof, and (2) increasing roof greening and distributing more energy as latent heat. Guo Liangchen et al. used the mesoscale numerical model WRF coupled with the urban single-layer canopy scheme (UCM) and took Nanjing as an example to simulate the mitigation effect of different high albedo roofs and different proportions of green roofs on urban high temperatures. The results showed that the high albedo roof with 0.8 and the 100% green roof had a similar cooling effect [34]. Detommaso, Gagliano et al. studied the influence of cold materials and urban forestation on the urban microclimate by a computational fluid-dynamic (CFD) model. The results show that a high level of urban greening is achieved through a large area of green roofs and urban greening, which ensures the health of pedestrians [29].

In this paper, we searched all the studies that applied LCZ data to the WRF model, and found that there are still three problems: (1) Since the input data required by the multi-layer urban canopy model are difficult to obtain, such as building energy consumption data, most of the studies adopt a single-layer urban canopy model, ignoring the indoor thermal environment of buildings. (2) An urban canopy model requires urban canopy parameters (UCPs) to determine the characteristics of buildings [35]. However, most of the studies directly use the default canopy parameters table for LCZ provided on the WUDAPT website, which is only applicable to 44 cities in the United States, but not to specific cities [36]. (3) In many studies, due to the difference of the urban area among different land use data, it is impossible to confirm whether the improvement of simulation accuracy in the model results is the effect of the LCZ refinement zone or urban area correction, so it cannot reflect the advantages of LCZ data. Therefore, to study the vertical structure and indoor thermal environment characteristics of Beijing in more detail, the multi-layer urban canopy scheme (BEM) considering the building energy model is selected. In order to analyze the improvement effect of LCZ on the model more accurately, the improved urban canopy parameter values are adopted. At the same time, this study designed two groups of different land use data set introduction models. In these two sets of updated data sets, under the condition of ensuring the same urban area, the schemes of no refined urban zone and a local climate zone are set up, respectively. By driving the model with the updated data set and the default MODIS data set, respectively, and comparing the simulation results with the measured values, the improvement of the model simulation effect by different land use data can be analyzed.

## 2. Data and Methods

### 2.1. Study Area

Beijing is located in the north of North China Plain, which is 39°28′ N–41°05′ N, 115°20′ E–117°30′ E, the center is 39°54′20″ N, 116°25′29″ E, with a total area of 16,410.54 km$^2$. Beijing is surrounded by mountains in the northwest and plains in the southeast. The urban area is concentrated in the plain area. The climate of Beijing is warm temperate, semi-humid and semi-arid. It is hot and rainy in summer, cold and dry in winter, short in spring and autumn, and long in winter and summer, with sufficient sunshine. The landscape pattern of Beijing is diverse, from the ancient open low-rise buildings to the modern style compact high-rise skyscrapers, which makes it very suitable for testing the improvement of numerical model simulation by fine land use classification. Beijing contains 18 meteorological stations, including Shunyi (SY), Haidian (HD), Yanqing (YQ), Miyun (MY), Pinggu (PG), Chaoyang (CY), Changping (CP), Mengtougou (MTG), Beijing (BJ), Shijingshan (SJS), Fengtai (FT), Daxing (DX), Fangshan (FS), Huairou (HR), Shangdianzi (SDZ), Tongzhou (TZ), Zhaitang (ZT), Xiayunling (XYL). At the same time, some studies have pointed out that the proportion of different urban functional areas has a very high regional dependence. The characteristics of cities in the same urban agglomeration are similar to each other [37]. Beijing is the representative of the Beijing–Tianjin–Hebei Urban Agglomeration. Taking Beijing as an example, the results can also be used as a reference for other Beijing–Tianjin–Hebei cities. Therefore, this paper selects Beijing as the research area.

## 2.2. Data

In this paper, the Landsat8 satellite data products with a resolution of 30 m provided by computer network information center Geospatial Data Cloud Platform (http://www.gscloud.cn, accessed on 10 April 2022) were used to generate the land use data set of Beijing LCZ, and the data of different seasons with cloud cover of less than 10% at the imaging moment were screened out to avoid inaccurate classification caused by vegetation cover change.

When running the WRF model, Final Operational Global Analysis data (FNL) provided by the National Centers for Environmental Prediction (NCEP) in the United States, with a spatial horizontal resolution of $1° \times 1°$ and a temporal resolution of 6 h, is used as the initial field and boundary field data of the driving model.

The measured data come from the hourly ground observation data set of China Meteorological Data Network (http://data.cma.cn/site/index.html, accessed on 10 April 2022). This paper utilizes the observation data of 18 ground weather stations in Beijing.

## 2.3. Design of Numerical Test Scheme

In this paper, the WRF model (version 4.3.3) is used to simulate the summer temperature in Beijing. The simulation time of this experiment is from 12:00 on 28 August 2019 to 00:00 on 1 September 2019, in which the first 12 h are the spin-up time. During this period, there was no cloud over Beijing and the weather was fine. The WRF test set up three levels of nesting Figure 1a, and the innermost region contains the entire Beijing city. The horizontal resolution of the grid is 27 km, 9 km, and 3 km, respectively, and the number of grids is $100 \times 104$, $88 \times 97$, and $97 \times 79$, respectively. The model is divided into 35 layers in the vertical direction and the top layer is 50 h pa. Other physical parameterization scheme settings are shown in Table 1. In this study, the observation data of 18 meteorological observation stations in the Beijing Administrative Region were selected to verify the simulation results, and the simulation effects of three groups of numerical experiments on 2 m temperature were compared and analyzed. The distribution of observation stations is shown in Figure 1b. The LCZ map of Beijing is shown in Figure 1c. The urban underlying surface is divided into 15 types, including nine built types (LCZ 1–9) and six land cover types (LCZ A, B, D, E, F, G).

Table 1 shows the physical parameterization scheme selected in this paper. The WRF parameterization scheme included the Noah land surface model [20], revised MM5 surface layer scheme [38,39], WSM6 microphysics scheme [40], RRTM long-wave radiation scheme [41], Dudhia short-wave radiation scheme [42], and Kain–Fritsch cumulus parameterization scheme [43] for the outermost domain. The Yonsei University (YSU) planetary boundary layer scheme [44] was also used. The above seven physical parameterization schemes were selected in this study because a previous work indicated that they can better simulate the urban thermal environment characteristics of Beijing during the simulation period [45].

**Table 1.** Main physical parameterization scheme settings.

|  | d01 | d02 | d03 |
|---|---|---|---|
| Microphysical processes | Wsm6 simple ice scheme | Wsm6 simple ice scheme | Wsm6 simple ice scheme |
| Long-wave radiation | RRTM | RRTM | RRTM |
| Short-wave radiation | Dudhia | Dudhia | Dudhia |
| Near surface layer | Revised MM5 | Revised MM5 | Revised MM5 |
| Land surface process | Noah scheme | Noah scheme | Noah scheme |
| boundary layer | Ysu scheme | Ysu scheme | Ysu scheme |
| Cumulus parameterization | Kain–Fritsch | Kain–Fritsch | Kain–Fritsch |

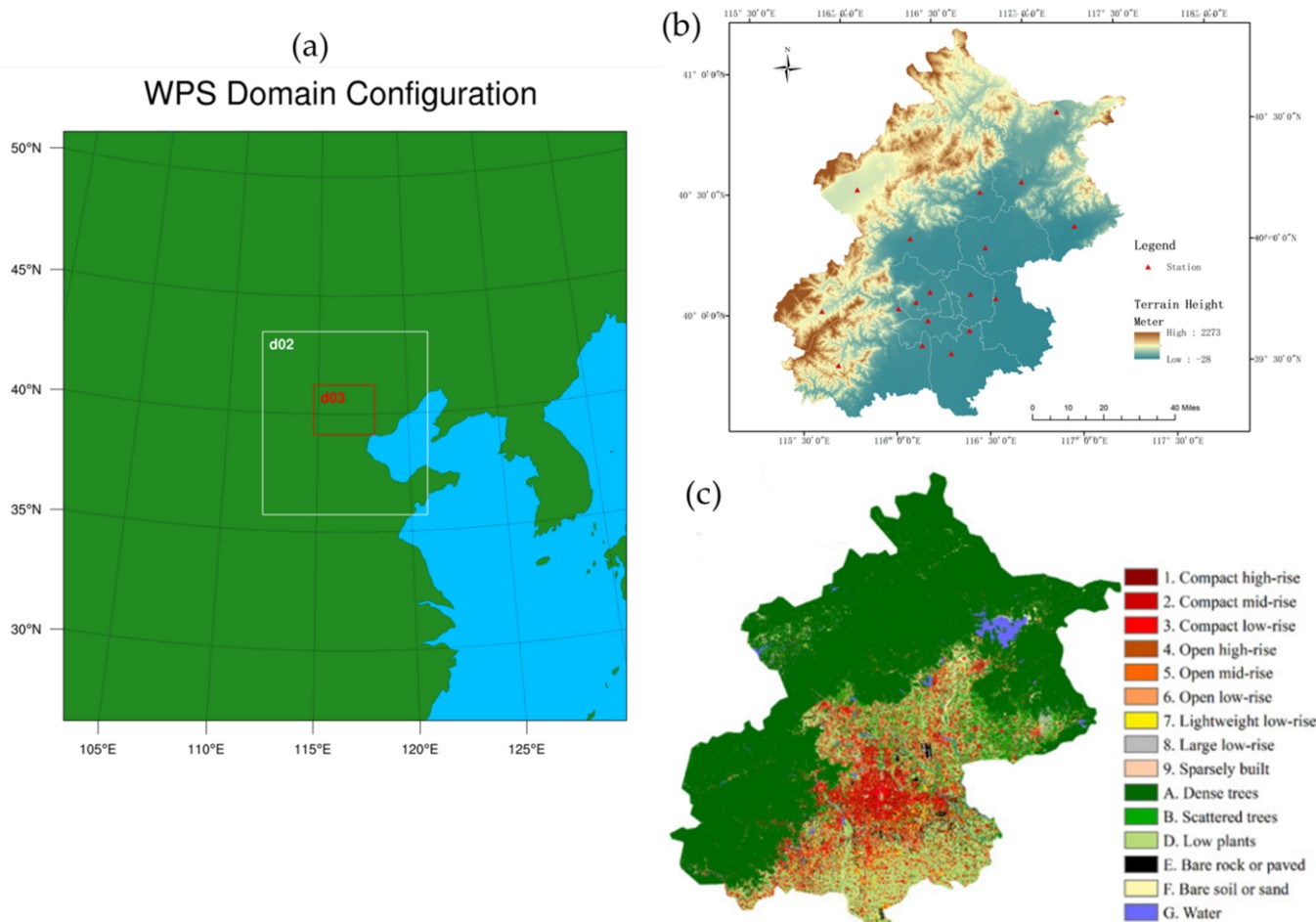

**Figure 1.** (**a**) Layout of nested areas, (**b**) distribution of meteorological stations. The red dot indicates the location of the weather station, (**c**) LCZ map of Beijing. Numbers 1–9 represent built types. A–G represent land cover types.

In this paper, three numerical cases (Case 1–Case 3) are set up for the simulation test, as shown in Table 2. The design of the cases is simple to complex, and the settings of each scheme are as follows: 1. Case 1 indicates that the default MODIS land cover data are used as the land use data of the underlying surface for numerical simulation. In this case, the urban area is small, and there is only one classification representing the urban area. As a control scheme, this case is compared with other land use data updating schemes to evaluate the improvement of land use data updating on model results. 2. Case 2 indicates that the data set of "only revise the urban area" is used as the land use data of the underlying surface for numerical simulation. In this data, the urban scope is larger than that of Case 1, and the urban area accounts for 64.84% of the total area of Beijing. In Beijing, there is only one land use category representing the urban area. Compared with Case 1, the updating of urban land use information, in this case, is only reflected in the expansion of the urban area. The purpose of this case is to study the improvement of the simulation results when only the urban area is updated, but the internal functional areas of the city are not finely divided. 3. Case 3 indicates that the LCZ data in the numerical experiment are used as the land use data of the underlying surface for numerical simulation. In this data, the urban area is larger than that of Case 1, accounting for 64.84% of the total area of Beijing. In Beijing, there are nine different local climate zones representing urban areas. Compared with Case 1, this case updates the urban land use information, which not only reflects the expansion of the urban area, but also reflects the refined zone within the city, which is closer to the actual situation. The purpose of this case is to study the improvement of urban

zoning on the simulation results. It should be noted that this paper only updates the land use data of urban areas, and the classification of the natural surface remains unchanged.

**Table 2.** Case name and underlying surface setting in Beijing.

| Case Name | Case 1 | Case 2 | Case 3 |
|---|---|---|---|
| Underlying surface information | MODIS scheme | Scheme of correcting only urban area | Local climate zone scheme |
| Number of urban internal classifications | 1 | 1 | 9 |
| Urban canopy model | BEM | BEM | BEM |

In this paper, the land use data of Case 2 and Case 3 are imported into WRF, as shown in Figure 2, which can intuitively show the differences of urban land use data in the three schemes.

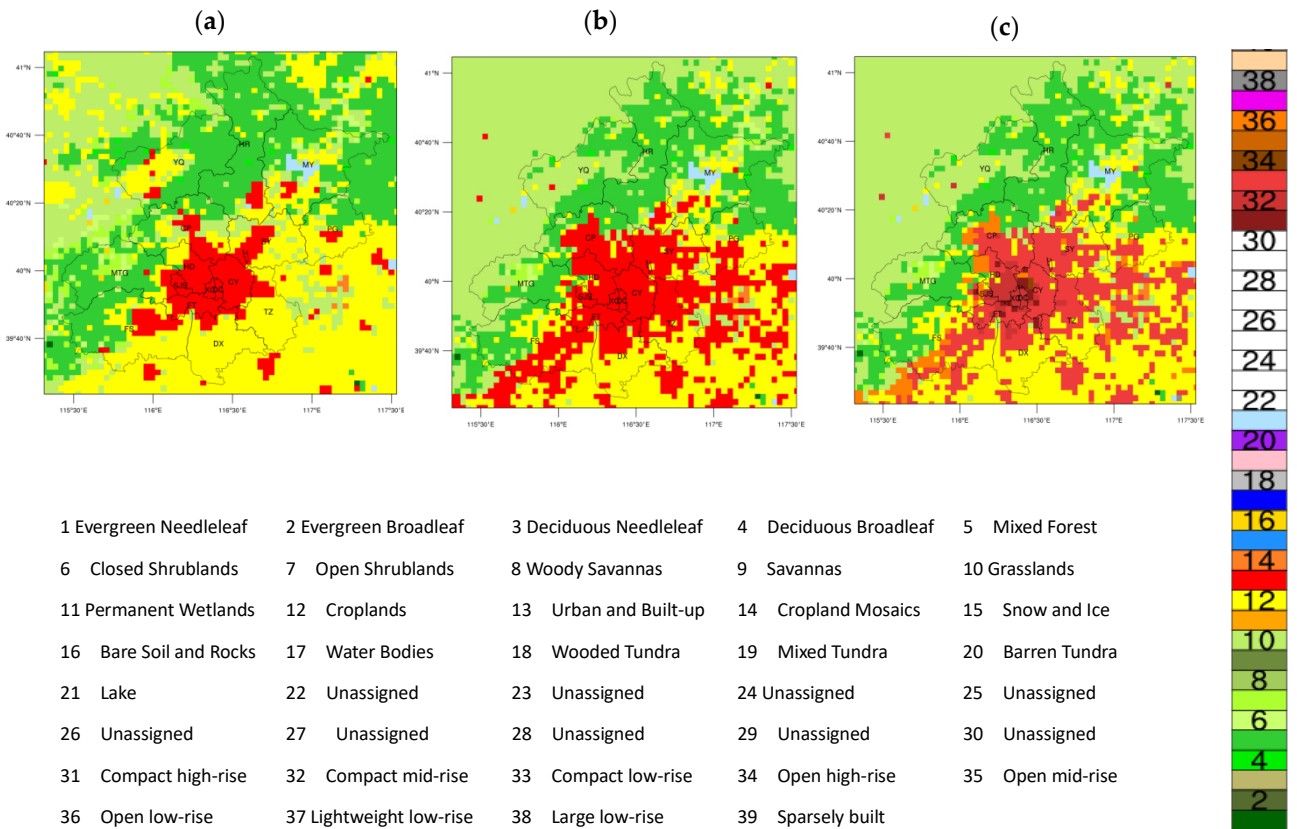

| 1 Evergreen Needleleaf | 2 Evergreen Broadleaf | 3 Deciduous Needleleaf | 4 Deciduous Broadleaf | 5 Mixed Forest |
|---|---|---|---|---|
| 6 Closed Shrublands | 7 Open Shrublands | 8 Woody Savannas | 9 Savannas | 10 Grasslands |
| 11 Permanent Wetlands | 12 Croplands | 13 Urban and Built-up | 14 Cropland Mosaics | 15 Snow and Ice |
| 16 Bare Soil and Rocks | 17 Water Bodies | 18 Wooded Tundra | 19 Mixed Tundra | 20 Barren Tundra |
| 21 Lake | 22 Unassigned | 23 Unassigned | 24 Unassigned | 25 Unassigned |
| 26 Unassigned | 27 Unassigned | 28 Unassigned | 29 Unassigned | 30 Unassigned |
| 31 Compact high-rise | 32 Compact mid-rise | 33 Compact low-rise | 34 Open high-rise | 35 Open mid-rise |
| 36 Open low-rise | 37 Lightweight low-rise | 38 Large low-rise | 39 Sparsely built | |

**Figure 2.** Main land use map. (**a**) Case 1, (**b**) Case 2, (**c**) Case 3.

In Table 3, according to the longitude and latitude information of the grid point closest to each station in the model, the main land use types of all measured stations in each Case are extracted.

**Table 3.** Main land use types of each station.

| Station ID | Station Name | Land Use Classification | | |
|---|---|---|---|---|
| | | Case 1 | Case 2 | Case 3 |
| UR1 | Shunyi | Urban and built-up | Urban and built-up | Open low-rise |
| UR2 | Haidian | Urban and built-up | Urban and built-up | Compact mid-rise |
| UR3 | Yanqing | Urban and built-up | Urban and built-up | Compact mid-rise |
| UR4 | Miyun | Urban and built-up | Urban and built-up | Open low-rise |
| UR5 | Pinggu | Urban and built-up | Urban and built-up | Compact mid-rise |
| UR6 | Chaoyang | Urban and built-up | Urban and built-up | Compact high-rise |
| UR7 | Changping | Urban and built-up | Urban and built-up | Compact mid-rise |
| UR8 | Mengtougou | Urban and built-up | Urban and built-up | Sparsely built |
| UR9 | Beijing | Urban and built-up | Urban and built-up | Compact mid-rise |
| UR10 | Shijingshan | Urban and built-up | Urban and built-up | Compact high-rise |
| UR11 | Fengtai | Urban and built-up | Urban and built-up | Compact mid-rise |
| UR12 | Daxing | Urban and built-up | Urban and built-up | Light-weight low-rise |
| UR13 | Fangshan | Urban and built-up | Urban and built-up | Compact mid-rise |
| RUR1 | Huairou | Woody savannas | Woody savannas | Woody savannas |
| RUR2 | Shangdianzi | Savannas | Savannas | Savannas |
| RUR3 | Tongzhou | Croplands | Croplands | Croplands |
| RUR4 | Zhaitang | Grasslands | Grasslands | Grasslands |
| RUR5 | Xiayunling | Savannas | Savannas | Savannas |

*2.4. Mode Evaluation Method*

In this paper, three commonly used statistical parameters are selected to evaluate the simulation results. These three parameters are correlation coefficient (R), mean absolute error (MAE), and root mean square error (RMSE). The formulae for each statistical parameter are as follows:

$$\text{R} = \frac{1}{N}\sum_{i=1}^{N}\left(F_i - \overline{F}\right)\left(O_i - \overline{O}\right) \Big/ \left( \sqrt{\frac{1}{N}\sum_{i=1}^{N}\left(F_i - \overline{F}\right)^2} \sqrt{\frac{1}{N}\sum_{i=1}^{N}\left(O_i - \overline{O}\right)^2} \right)$$

$$\text{RMSE} = \sqrt{\frac{1}{N}\sum_{i=1}^{N}\left(F_i - O_i\right)^2}$$

$$\text{MAE} = \frac{1}{N}\sum_{i=1}^{N}\left|F_i - O_i\right|$$

In the above calculation formula, $F_i$ represents the simulation value, $\overline{F}$ represents the average simulation result, $O_i$ represents the observation value of the station, $\overline{O}$ represents the average observed value of the station, and $N$ represents the total number of samples. In this paper, there are 18 stations and 72 times, so the total number of samples is 1296. MAE and RMSE are both indicators to measure the error between the model and the measured value. The smaller the error, the better the model effect.

*2.5. The Urban Heat Island Effect*

In heat island studies, researchers often artificially change the land use data in the model from urban to rural, explain the impact of urbanization by calculating the difference field between urban and non-urban examples, and define the different fields of temperature as urban heat island intensity (UHII) [46]. This method is in line with Lowry's idea that "to analyze the impact of cities on climate, the present situation should be compared with the original state before the appearance of cities", and the influence of land, sea, wind, topography, and clouds can be excluded [47]. Therefore, according to the above method, this paper designs a new example of "removing" cities: Case NoUrban. In this example, by modifying the main land use types and land use scores, the main land use type "farmland

and pasture" in Beijing in MODIS default data replaces the area classified as a city in the default data. In this paper, the hourly UHII of stations in each case in the simulation period is calculated, that is, the temperature difference from Case 1–3 with city and Case NoUrban without the city.

## 3. Results

### 3.1. Analysis of 2 m Daily Temperature Variation

The 2 m temperature refers to the surface air temperature (the temperature at a height of 2 m). The measuring environment of the 2 m temperature is as follows: An observation field is an area in which instruments are arranged in an efficient and appropriately concentrated manner. It should be level, open, flat, and not obscured by trees or buildings. On barren land, the ground should be covered with natural grass and should be enclosed by fences that do not prevent wind from passing through. The ground should be kept clean year-round by occasional mowing and weeding. Locations on steep slopes or depressions should be avoided because of the poor representation caused by this terrain. Power and water supplies for the observation field and instrument management and maintenance are beneficial (https://vdocument.in/cp2-temperature.html, accessed on 10 April 2022).

It can be seen from Figure 3 that the temperature curve illustrates a single peak distribution, with the lowest temperature around 5:00 a.m., and the temperature gradually rises after sunrise, reaching the peak at about 16:00. All cases can well simulate the daily variation trend of temperature. The simulation results of Case 1 for the 2 m temperature are always lower than the measured values. In the last two cases, by artificially updating the land use data, the simulation effect of the model on temperature has been improved to varying degrees. The results in Case 2 and Case 3 are close to the measured values. For the non-urban station (RUR1–5), the diurnal variation of the simulated temperature of the three groups of experiments is similar and the urban canopy scheme will not affect the non-urban areas, so the change of the underlying surface in urban areas has little impact on the simulation results of the natural surface. For the urban station (UR1–13), from the statistical results (Table 4), the simulated temperature of the Case 2 test is the closest to the observed data, with the correlation coefficient R of 0.93 and RMSE of 1.84 °C. The Case 2 results are better than the Case 1 results, indicating that the scheme of correcting only the urban area can better simulate the urban local climate characteristics than the MODIS scheme. The simulation results of the Case 3 test are closely related to the underlying surface types, such as the UR1 station. In the Case 3 test, the UR1 station is an open low-rise building area, and the simulation results for the 2 m temperature are closer to the observed data than the Case 1 and Case 2 test results. The main reason for the difference between the Case 2 and Case 3 test results is that the setting of the urban parameter table is different. The underlying surface of Case 3 is LCZ, which has different parameter settings for each city classification, so the temperature simulation results are closely related to the underlying surface types. In the Case 2 test, there is only one type of city classification and there is only one set of corresponding parameter tables. Although the simulation results are good, it cannot reflect the energy difference between building types.

**Table 4.** Statistical results of 2 m temperature.

|  | Case 1 | Case 2 | Case 3 |
|---|---|---|---|
| RMSE (°C) | 3.03 | 1.84 | 2.27 |
| MAE (°C) | 0.03 | 0.01 | 0.01 |
| R | 0.89 | 0.93 | 0.94 |

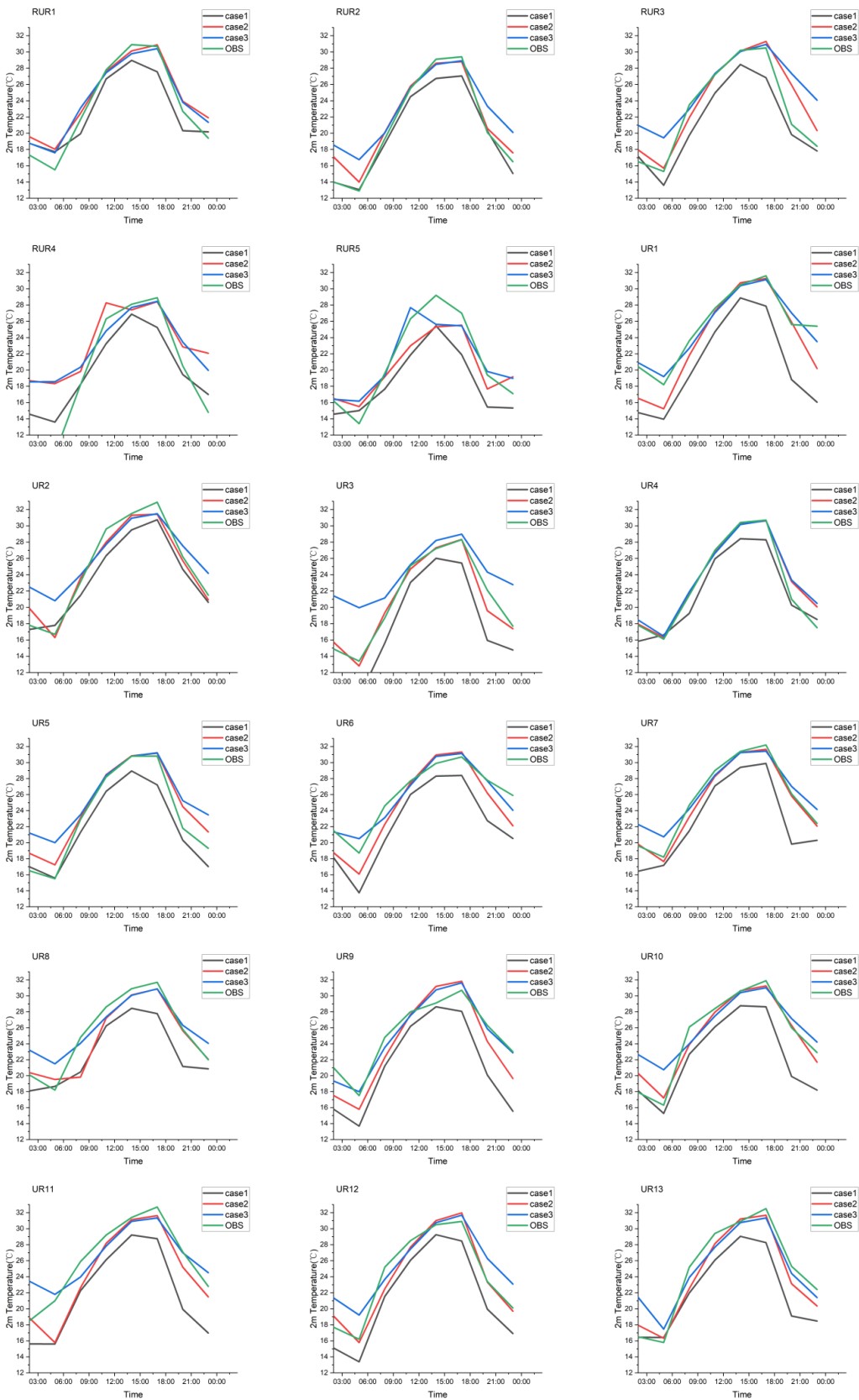

**Figure 3.** Comparison of three-day mean daily variations of 2 m temperature simulated and observed at 18 stations in Beijing (OBS: observed data from 18 weather stations in Beijing).

In order to more clearly illustrate the improvement effect of the updated underlying surface data on the model, this paper calculates the RMSE value of the temperature at each station in each case, and makes a box diagram, as shown in Figure 3. It can directly show the RMSE distribution of the temperature at each station in each case. As can be seen from Figure 4, the RMSE of Case 1 ranges from 1.35 °C (Shangdianzi station) to 5.33 °C (Shunyi station). The RMSE of Case 2 ranges from 0.66 °C (Changping station) to 4.88 °C (Zhaitang station), and the RMSE of Case 3 ranges from 1.02 °C (Shunyi station) to 4.59 °C (Zhaitang station). The maximum RMSE of Case 1 is in the Shunyi Station, however, the minimum RMSE of Case 3 is in the Shunyi Station. The reason for this may be that the Shunyi Station is an open low-rise building in Case 3, which is conducive to ventilation, so the simulated temperature of Case 3 is closer to the observed value than that of Case 1. The average RMSE of Case 1, Case 2, and Case 3 are 3.03 °C, 1.84 °C, and 2.27 °C, respectively. Compared with Case 1, the RMSE of Case 2 is reduced by 1.19 °C, which indicates the influence of the urban area correction on the simulation results. Compared with Case 2, the RMSE of Case 3 is increased by 0.43 °C, which indicates the influence of refined classification within the city on the simulation results. The RMSE of Case 2 has an outlier (4.88 °C) in the Zhaitang Station.

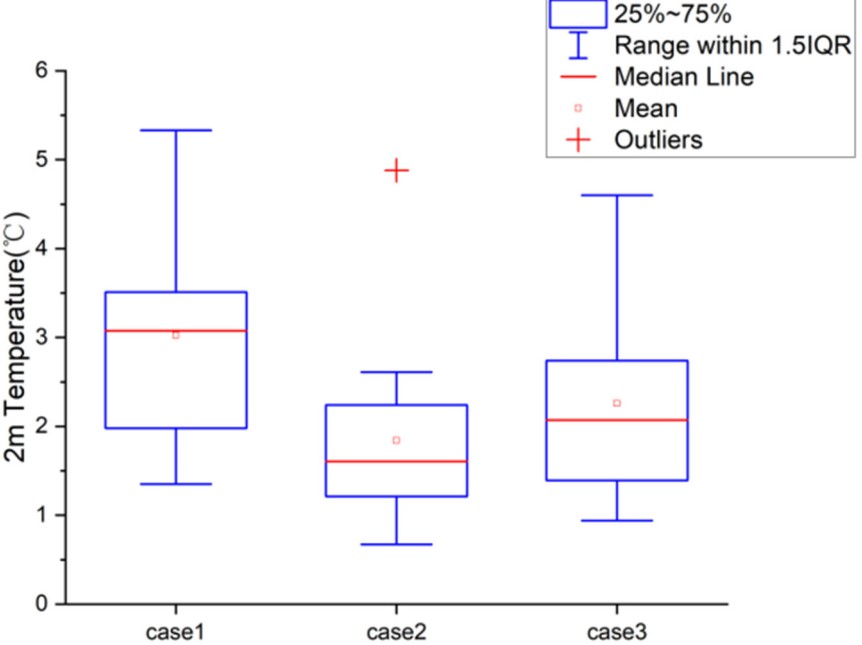

**Figure 4.** RMSE distribution of 2 m temperature at stations.

### 3.2. Spatial Distribution Analysis of Thermal Environment

3.2.1. Spatial Distribution Analysis of 2 m Temperature

In the previous section, through the statistical analysis of the temperature simulation results, it is found that the thermal environment is greatly affected by the underlying surface, so this section will analyze the spatial distribution of 2 m temperature in the simulation results. From the daily variation curve of the temperature, it can be seen that in the simulation period, the daily minimum and maximum temperatures appear at around 05:00 local time (LT) and 16:00 LT, respectively. Therefore, in this section, the average temperatures of each case at 05:00 and 16:00 are calculated, and the spatial distribution map of the 2 m temperature is drawn.

Figures 5 and 6 show the 2 m temperature distribution of the model simulation results at 05:00 and 16:00. In all the simulation results, the temperature in the urban area is 2–4 °C higher than that in the suburbs, and the low temperature is distributed in the northwest region, which is consistent with the surface morphological characteristics of

Beijing, which is surrounded by mountains in the northwest and urban agglomeration areas in the southeast.

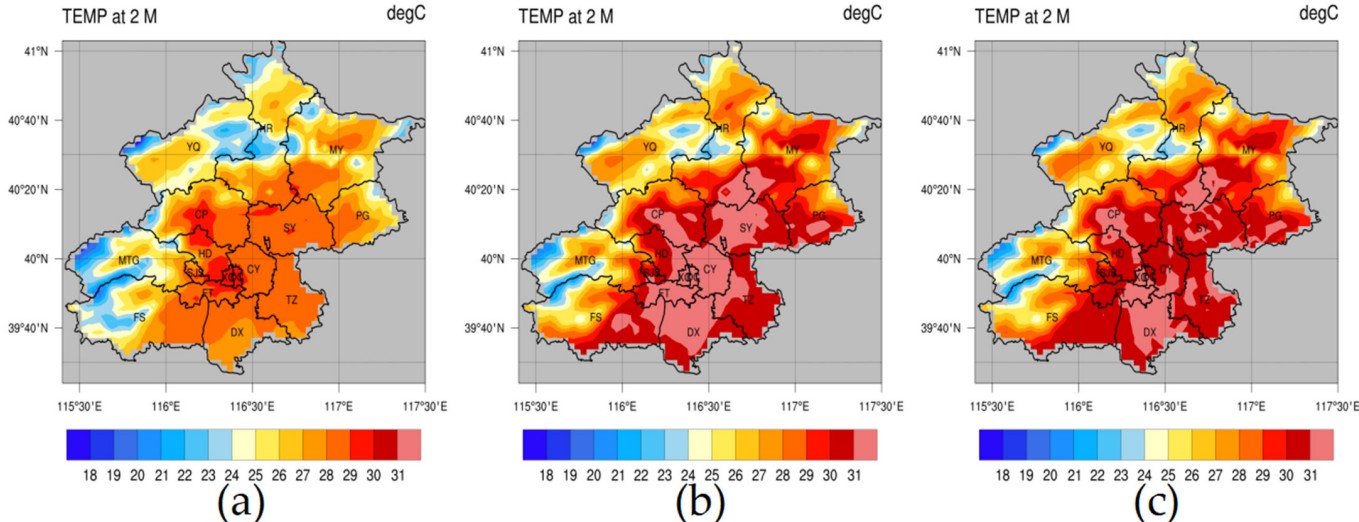

**Figure 5.** The 2 m temperature distribution at 05: 00 (°C). (**a**) Case 1 simulation value, (**b**) Case 2 simulation value, (**c**) Case 3 simulation value.

**Figure 6.** The 2 m temperature distribution at 16: 00 (°C). (**a**) Case 1 simulation value, (**b**) Case 2 simulation value, (**c**) Case 3 simulation value.

As shown in Figure 5, in all the simulation results before sunrise, the surface temperature in the urban area is higher than that in the suburban area, and there is an obvious urban heat island phenomenon. The results show that the simulated urban temperature of Case 2 is similar to that of Case 1, and the overall temperature in Case 3 is on the high side, which has a big error with the actual measurement, but the spatial distribution of temperature is more uniform and reasonable. As shown in Figure 6, in Case 1, the temperature of no or few areas exceeds 30 °C, while in Cases 2 and 3, there are obvious high-temperature areas. In Case 3, because each LCZ type has different thermal characteristics, the temperature distribution in the urban area is uneven. The average temperature of LCZ 1, LCZ2, LCZ3, LCZ4, LCZ5 LCZ6, and LCZ8 in the urban center is about 32 °C, while the average temperature of LCZ 7 and LCZ 8 in the northwest suburb of the city is only about 25 °C under the influence of lake wind. The LCZ8 exhibits different temperature characteristics at different locations, so we infer that the same LCZ type shows different microclimate

characteristics due to their being located in different urban locations, which indicates that the thermal environment inside the city depends not only on the land use type, but also on the interaction between the local area and the atmospheric circulation.

As shown in Figure 6, because the urban scope of Case 2 is larger than that of Case 1, the simulated high-temperature space scope is larger than that of Case 1. As can be seen from Table 4, the temperature simulated by Case 2 is closest to the data of the observation station, so the temperature in Case 1 is greatly underestimated due to the small city area. The urban scope of Case 3 is larger than that of Case 1, so the simulated high-temperature space scope is larger than that of Case 1. At the same time, there is only one category in Case 1, while the cities in Case 3 are subdivided into nine categories. Compared with the two cases, it can be seen that the temperature of Case 3 is spatially heterogeneous in the urban area, and the temperature of Case 1 is evenly distributed in the urban area.

### 3.2.2. Spatial Distribution Analysis of Surface Heat Flux

The spatial distribution of sensible heat flux and latent heat flux at 16:00 is shown in Figures 7 and 8, respectively. The sensible heat flux mainly reflects the energy exchange between the earth and the atmosphere, which is related to the difference between the surface temperature and the near-surface temperature. The greater the temperature difference between the earth and the atmosphere, the greater the sensible heat flux. The latent heat flux reflects the water vapor exchange between the earth and the atmosphere, which is mainly related to the evapotranspiration of water vapor from the surface to the atmosphere. Generally speaking, the soil and vegetation on the surface of the suburbs contain a lot of water, and the latent heat flux is large, while the urban surface is a mainly impervious surface, so the latent heat flux is small. As can be seen from Figure 7, on the whole, the maximum sensible heat fluxes of the three cases are located in urban areas. In Case 1, the range greater than 125 W/m$^2$ is small, mainly in the Dongcheng District and Xicheng District. The spatial distribution of Case 2 and Case 3 is similar. Case 2 has a large range of sensible heat values greater than 160 W/m$^2$, the sensible heat flux of a sparse building area (LCZ9) is the highest in Case 3, and the sensible heat flux of an open middle-rise building area (LCZ5) is the lowest. The spatial range of the high-value of the sensible heat flux in Case 2 is larger than that in Case 1, and the underestimation of the urban area in Case 1 also causes the high-value area of the sensible heat flux in Case 1's simulation results to be too small. The high-sensitivity hot zones of Case 3 and Case 1 are located in the Dongcheng District and Xicheng District of Beijing, which is also the location of the core area of Beijing. However, compared with Case 1, the sensible heat distribution in Case 3 is more reasonable, and the sensible heat value decreases slowly as the built-up area spreads around the core functional area.

As can be seen from Figure 8, the latent heat in the city is small and the latent heat in the suburbs is large. The default MODIS land use data underestimate the urban area, which also causes the low-value area of the latent heat flux in Case 1's simulation results to be too small. The latent heat flux difference between city and suburb in Case 2 is obvious, with the boundary between the city and suburb generally reaching more than 200 W/m$^2$. This situation of an excessive latent heat flux gradient does not conform to the law that land use in the urban fringe gradually transits to the suburbs. In Case 3, the latent heat distribution is more reasonable. The low latent heat area is located in the Xicheng and Dongcheng areas, which is also the location of the core area of Beijing, and with the built-up area spreading out around the Xicheng and Dongcheng District as the center, the latent heat value increases slowly. Since the city scope of Case 2 is larger than that of Case 1, the spatial range of the low value of the latent heat flux in the simulation results of Case 2 is larger than that of Case 1. The Case 3 and Case 1 low latent heat zones are located in the Xicheng District and Dongcheng District of Beijing, which is also the location of the core area of Beijing. In Case 1, the latent heat flux between urban and suburban areas is obviously different. However, the latent heat distribution in Case 3 is more reasonable, and the latent heat value increases slowly as the built-up area spreads around the core area.

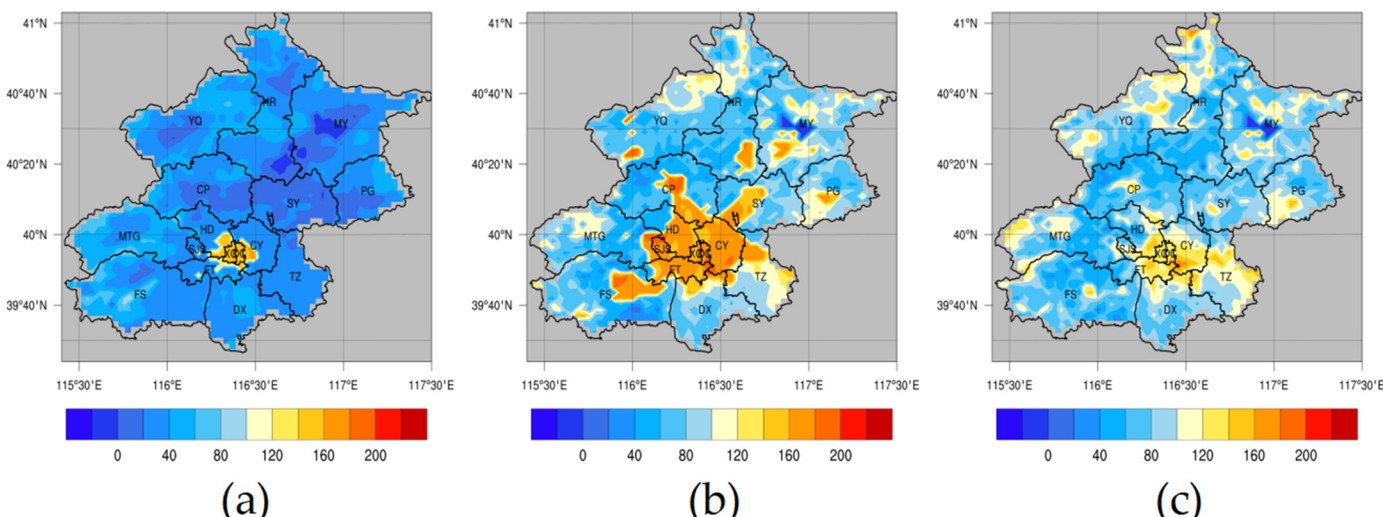

**Figure 7.** Distribution of sensible heat flux (W/m$^2$) at 16:00 (**a**) Case 1 simulation value, (**b**) Case 2 simulation value, (**c**) Case 3 simulation value.

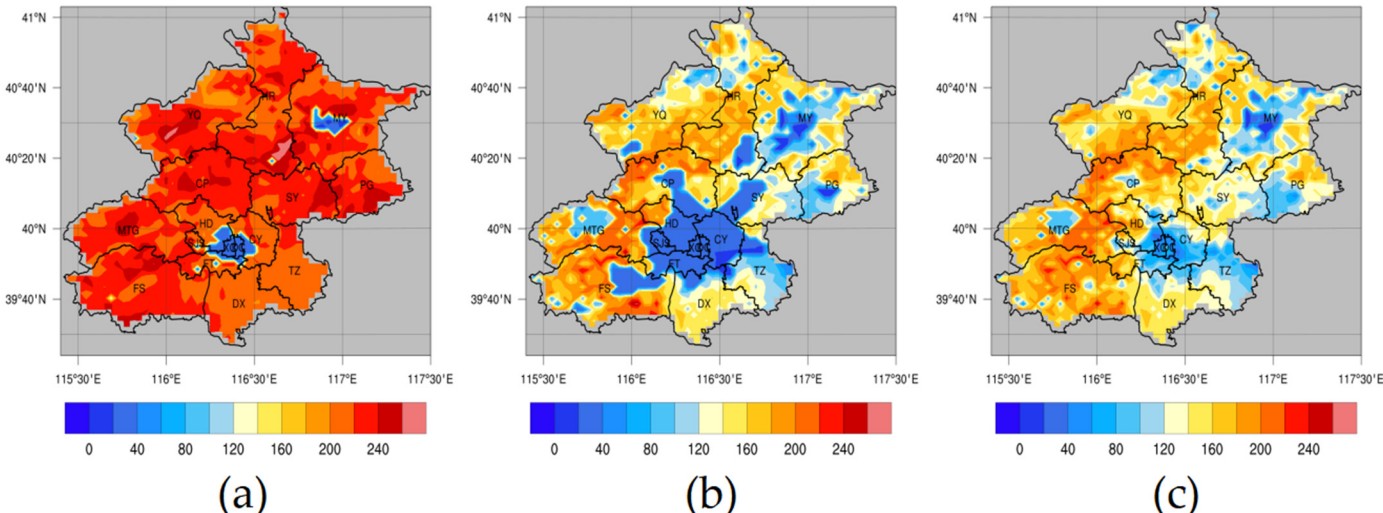

**Figure 8.** Distribution of surface latent heat flux (W/m$^2$) at 16:00 (**a**) Case 1 simulation value, (**b**) Case 2 simulation value, (**c**) Case 3 simulation value.

### 3.2.3. Heat Island Intensity Distribution

This section designs a new example of "removing" cities, Case NoUrban. In this case, the main land use type of Beijing in MODIS default data "farmland and pasture" replaces the areas classified as cities in the default data. In this section, the hourly UHII of stations in each case in the simulation period is calculated, that is, for the temperature difference between Case 1 and 3 with the city and Case NoUrban without the city, the histogram of the UHII daily variation is drawn, as shown in Figure 9.

It can be seen from Figure 9 that the Case 1 test underestimates the intensity of the heat island to a large extent. Except for the fact that the Haidian station has a heat island of no more than 2 °C before sunrise and after sunset, there is no obvious heat island phenomenon at other stations. In the two land use data updating schemes, except the Miyun station and Huairou station, an obvious heat island effect appeared and the diurnal variation pattern was the same, demonstrating the characteristics of low intensity in the daytime and high intensity in the night, and the minimum heat island intensity appeared at around 11:00, while the maximum intensity appeared at around 05:00. The diurnal variation of the heat island intensity in Case 3 was larger than that in Case 2.

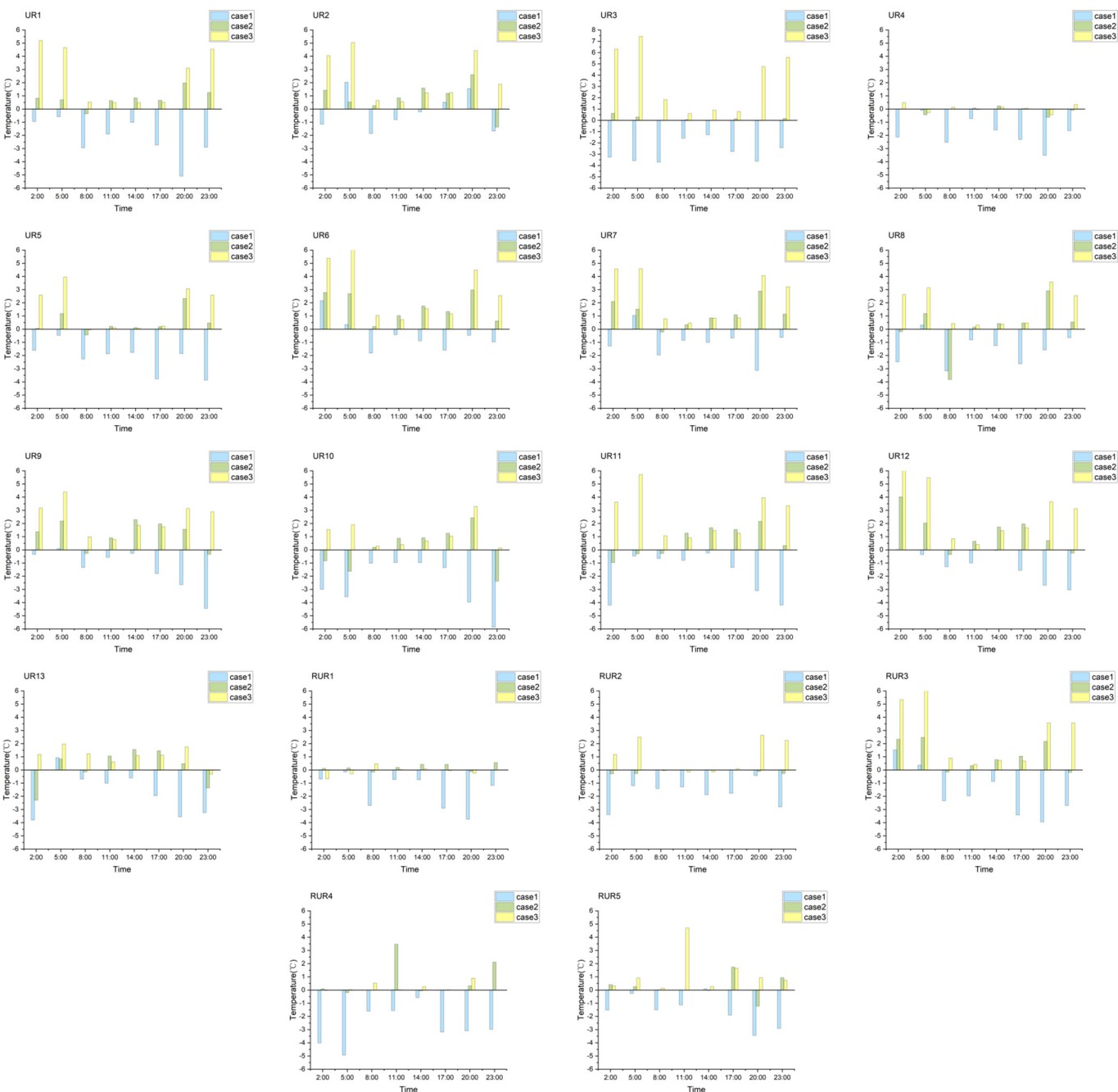

**Figure 9.** Diurnal variation of UHII at 18 stations in Beijing. (UR1: SY, UR2: HD, UR3: YQ, UR4: MY, UR5: PG, UR6: CY, UR7: CP, UR8: MTG, UR9: BJ; UR10: SJS, UR11: FT, UR12: DX, UR13: FS, RUR1: HR, RUR2: SDZ, RUR3: TZ, RUR4: ZT, RUR5: XYL).

Figures 10 and 11 show the spatial distribution of the heat island at 11 a.m. and 5 a.m., respectively. It can also be seen from the spatial distribution that the intensity of the heat island at night is greater than that in the daytime. At 11:00, the urban land coverage in Case 1 is extremely small, so the warming effect of the city is not obvious, so there is almost no heat island in Figure 10a. The heat island intensity of Case 2 is significantly higher than that of Case 3, but the spatial range of the heat island in Case 3 is larger than that in Case 2. At night, the central position of the heat island in Case 1 is roughly coincident with the urban land position in the MODIS land use data, but the heat island effect is weak and the intensity is less than 3 °C. However, in the two Cases with updated land use data, a heat island above 4 °C exists and the heat island intensity of Case 3 is higher than that of

Case 2. In Case 3, the effect of the urban internal zone difference on heat islands is obvious. The high-temperature heat island area is mainly concentrated in the Dongcheng District, Xicheng District, and other urban core functional areas.

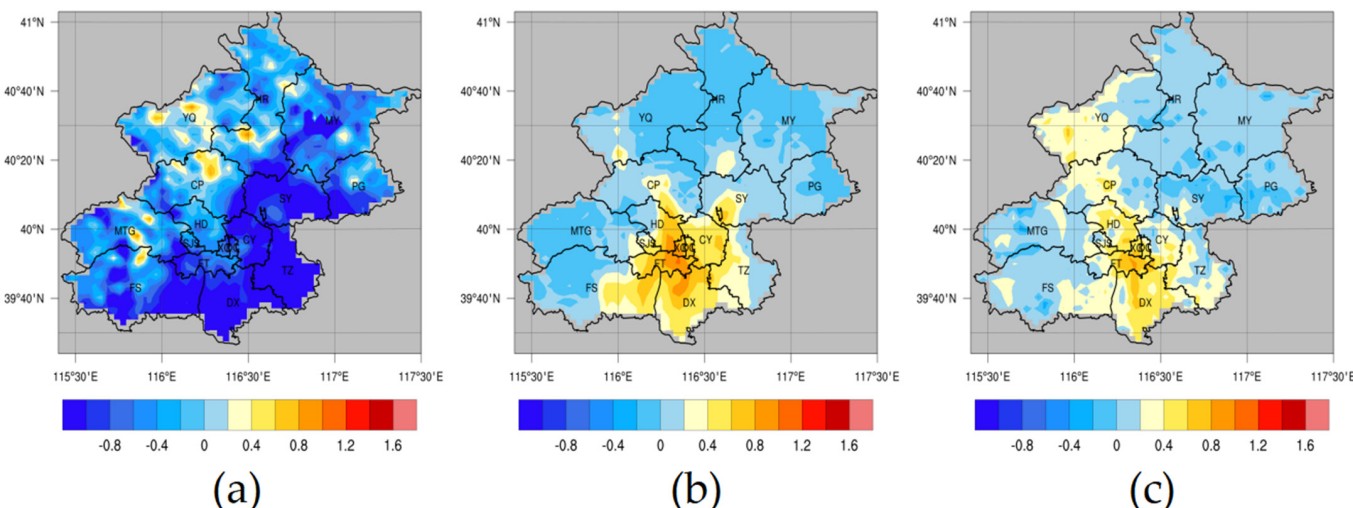

**Figure 10.** Surface heat island intensity distribution at 11:00 a.m. (°C) (**a**) Case 1–Case NoUrban, (**b**) Case 2–Case NoUrban, (**c**) Case 3–Case NoUrban.

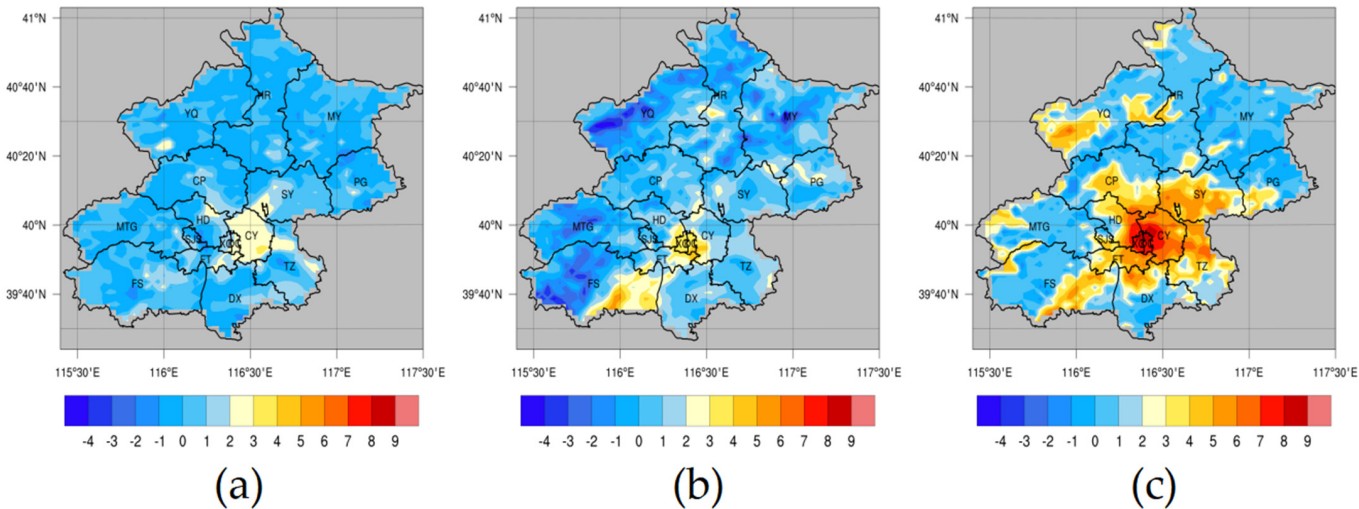

**Figure 11.** Surface heat island intensity distribution at 05:00 a.m. (°C) (**a**) Case 1–Case NoUrban, (**b**) Case 2–Case NoUrban, (**c**) Case 3–Case NoUrban.

*3.3. Influence of Urbanization Development on 2 m Temperature in Beijing*

As Case 1 uses the default MODIS data, this section only considers the impact of urbanization development on Beijing's temperature in the past 20 years (2000–2019) in Case 2 and Case 3. The simulation period is from 00:00, on 30 June 2019, to 00:00 on 1 August 2019. A previous study depicted that it was mostly sunny in Beijing in July 2019 [45], which is suitable for our research on the influence of the fine-scale underlying surface on the urban thermal environment, ss cloudy or rainy weather may increase the uncertainty of a simulated radiation flux and surface meteorological elements due to complex microphysical processes.

It can be seen from Figure 12 that the temperature difference of 2 m in July between Case 2 and Case 3 is between ±1.5 °C. The temperature difference in Case 2 is scattered in the urban area, while the temperature difference in Case 3 is continuous in the urban area. Moreover, the average temperature of the Beijing urban area in July 2019 is higher

than that in July 2000, which may be caused by the change of the land use structure from a natural surface to an urban surface between 2000 and 2019. Compared with Case 2, Case 3, which divides the city into nine categories, can better simulate the characteristics of local temperature differences in urban areas.

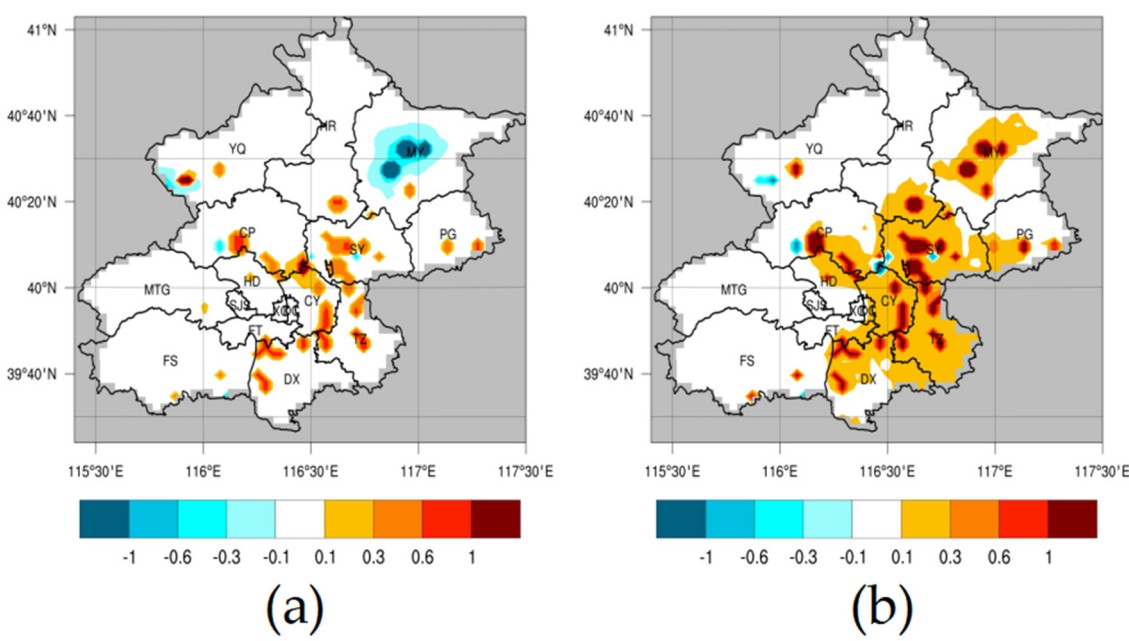

**Figure 12.** Spatial distribution of the average difference of 2 m temperature in July. (**a**) Case 2 simulation value, (**b**) Case 3 simulation value.

## 4. Discussion

In order to evaluate whether the simulation accuracy in the model results is improved by an LCZ refinement zone or urban area correction, the land use data without a refined urban zone and local climate zone were imported into the WRF-BEM model to simulate the 2 m temperature in Beijing under the condition of ensuring the same urban area. The results show that urban area correction significantly improves the simulation effect of the model on the 2 m temperature and the effect of the refined zone is small. In terms of a sensible heat flux and latent heat flux, a sensible heat value has an obvious high-value center in the area correction scheme, and the latent heat value gradient at the urban–rural boundary is too large, resulting in an unreasonable mutation in space. In the LCZ classification scheme, there is no obvious high-value center of sensible heat, and the latent heat distribution is more reasonable. As the built-up area spreads around the core functional area, the latent heat value increases slowly. At the same time, there are urban heat islands in Beijing day and night, and the intensity at night is much higher than that in the daytime. The total urban area in land use data affects the intensity and distribution of heat islands, and the refined zone within the city also strengthens the intensity of urban heat islands. Finally, the refined zone within the city can better reflect the impact of urbanization on the 2 m temperature.

Mu et al. (2020) discussed the influence of an LCZ map and underlying default-mode surface on the simulation capability by taking Beijing as an example [15]. The research proved that the LCZ map could improve the simulation effect of the model on the 2 m temperature, and the coupling scheme of LCZ-SLUCM was better than LCZ-BEM in the temperature simulation. In addition, their research adopted the default canopy parameterization setting; Liang et al. (2021) added an LCZ to the WRF-SLUCM model to simulate the 2 m temperature in Beijing and improved the urban canopy parameters locally. The results showed that improving the urban canopy parameters can significantly improve the simulation performance of the model [48]. In this study, an LCZ was combined with

the WRF-BEM model, and the improved urban canopy parameters were used to simulate the 2 m temperature. Compared with the results of Liang et al. (2021), the R increased by 0.14, and the RMSE decreased by 1.92 °C. To some extent, it can be concluded that the simulation results of the multi-layer urban canopy model (BEM) are better than that of single-layer urban canopy model (SLUCM), which is inconsistent with the conclusion of Mu et al. (2020), and it is worthy of further study in the future. Hu et al. (2020) used the WRF-SLUCM model to study the advantages of an LCZ. The results showed that the urban area modification significantly improved the simulation effect of the 2 m temperature, with a 0.77 reduction in the RMSE, while the fine zone had little effect with a 0.14 reduction in the RMSE [49]. In this study, the WRF-BEM model is used to study the advantages of LCZ data. The results show that the average RMSE of the temperature in the simulation decreases by 1.19 °C and the average R increases by 0.04 when only the urban area is modified. The refined classification of cities also affects the simulation results of temperature, but to a lesser extent, and the R is only increased by 0.01. Cai et al. (2021) showed that the high-temperature heat island area in Beijing is mainly concentrated in the compact low-rise (LCZ3), compact mid-rise (LCZ2), and large low-rise (LCZ8) types [50]. These three types of buildings are mainly located in the Dongcheng District and Xicheng District of the Second Ring Road in Beijing, which is consistent with our research results.

This study provides more possibilities for using the WRF-BEM model to test the improvement degree of the LCZ map on the simulation results, and the test results support further research on this WRF-BEM model. In this paper, the WRF model is used in the simulation study, the simulation period is short, and the conclusion has certain limitations. Therefore, the follow-up study will further evaluate and analyze the long-term simulation effect more comprehensively.

## 5. Conclusions

In this paper, two kinds of new land use data are generated by using open source satellite images and remote sensing methods. Combined with the default MODIS data of WRF, three cases are set up. Taking 0:00 on 29 August to 0:00 on 1 September 2019 as the research interval, using the WRF–BEM model, the effects of three Cases on the 2 m temperature, surface heat flux, and heat island effect in Beijing were simulated. After that, a month from 00:00 on 30 June 2019 to 00:00 on 1 August 2019 is selected to simulate and analyze the impact of the urban underlying surface changes in 2000 and 2019 on the 2 m temperature in Beijing. The following conclusions are obtained:

1.  Our research results show that the simulated 2 m temperature of the scheme of correcting only urban areas is the closest to the observed data, with an R of 0.93, an RMSE of 1.84 °C, and an MAE of 0.01 °C. There is a close relationship between the simulated results of the LCZ scheme and the types of the underlying surface. For example, the UR1 station is classified as open low-rise built, and the simulated result of the 2 m temperature is closer to the observed data than the MODIS scheme and scheme of correcting only urban areas.

2.  The MODIS scheme, the scheme of correcting only urban areas, and the LCZ scheme can simulate the diurnal variation characteristics of temperature. Although the RMSE in the 2 m temperature simulated by the LCZ scheme is 0.43 °C higher than that of the scheme of correcting only urban areas, it can well reproduce the spatial variation characteristics of the 2 m temperature.

3.  For the simulation of the surface heat flux, the sensible heat value of the scheme of correcting only urban areas has an obvious high-value center, and the latent heat value gradient is too large at the boundary between urban and rural areas, which forms an unreasonable mutation in the space. In the LCZ scheme, the sensible heat has no obvious high-value center, showing a linear band distribution of increasing sensible heat from the northwest to southeast, and the latent heat distribution is more reasonable. The low latent heat area is located in the Xicheng and Dongcheng areas, which is also the location of the core area of Beijing, and with the built-up area

spreading out around the Xicheng and Dongcheng District as the center, the latent heat value increases slowly.

4. Urban heat islands exist day and night in Beijing, and the intensity at night is much higher than that in the daytime. The total urban area in land use data affects the intensity and distribution of the heat island, and the difference in the urban internal division has a significant impact on the heat island. High-temperature heat island areas are mainly concentrated in compact low-level, compact mid-level, and large low-level types.

5. In the study of the impact of urbanization on the 2 m temperature, the LCZ scheme can more clearly reflect the temperature difference within urban areas.

**Author Contributions:** Initial idea and research plan were provided by X.Z. All authors contributed to the study conception and design. Material preparation, data collection, and analysis were performed by F.H., J.L. and Y.Z. The first draft of the manuscript was written by F.H. and all authors commented on previous versions of the manuscript. F.H. and M.Z. contributed more to the revision of the manuscript. All authors have read and agreed to the published version of the manuscript.

**Funding:** This research was funded by the National Natural Science Foundation of China (No. 72033005) and the Technology Research Project of Information Center, Ministry of Natural Resources (No. 2021-74-B2001-03).

**Institutional Review Board Statement:** Not applicable.

**Informed Consent Statement:** Not applicable.

**Data Availability Statement:** Data are available for use upon request.

**Acknowledgments:** We thank the Beijing Meteorological Informational Center for help with the meteorological data from the Beijing Meteorological Monitoring Stations.

**Conflicts of Interest:** The authors declare no conflict of interest.

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
