# Peer review of "Study on Urban Thermal Environment in Beijing Based on Local Climate Zone Method"

_sustainability, doi:10.3390/su14159503_

Round 1

Reviewer 1 Report

The paper presents a study on thermal analysis using different datasets and comparing simulation results. The study is conducted in Beijing, where urbanization is very high. The paper summarizes the relevant review in the introduction. The methodology part presents the applied approach. The results are supported by figures and statistical outputs. Most of the figures should be improved. The authors should also consider the questions and suggestions written below;

1.     The abbreviations should be given in a long version when they are written in the first place. In the abstract, please write the longer version of WRF and WFR-BEM.

2.     Lines 29-30: The sentence should be revised; it should not be in the future tense.

3.     Line 119: comma should be a dot.

4.     Figure 1 should be enhanced; it is not readable.

5.     It does not seem fair to compare three cases in which two of them have only one urban class. It can be expected that case 3 is going to provide different results as mentioned in the paper.

6.     What is “2m temperature” and how is the measurement environment? Please also explain the “actual observation”.

7.     Figure 3: What is OBS in the figure? It should be given in the caption with parenthesis.

8.     Figures 5 and 6: the caption starts with “m”. Is it 2m?

9.     Table 4 unit of the values should be added.

10.   Line 210: the word “Cases” should be lowercase.

11.   Lines 272-273: The LCZs are mentioned in these lines, however, no map shows these LCZ types. It should be added to indicate their temperature.

12.   The sentence starts in line 275 uses future tense. When we will see that different microclimate characteristic? Clarify the sentence.

13.   Check all captions of the figures and tables. They should start capital.

14.   Figures between 5 and 12: the location of the urban is not clear. They should be larger and the borders and letters should be bigger.

15.   Page 12: write the abbreviations of the stations in the manuscript when explaining them. It is difficult to follow between the text and figure 9.

16.   Line 336: Fig.5 should be Fig.10?

17.   On page 23 why values on July are given is not clear? Is there a specific reason for that month? That should be mentioned in the manuscript.

18.   Line 392: write the unit of the RMSE value.

19.   The results indicate that the improvement of R value and temperature is quite low (0,04 R and 1.16 C). How important having those limited levels of improvement? Is it significant to improve these levels?

Reviewer 2 Report

Line 9: correct: “Correspondence: Correspondence:”

Line 11: “WRF”, mention full term for the first use.

“Figure 5. m temperature distribution ..., Figure 6. m temperature distribution... “: do you mean 2m temperature?

Line 312: “3.2.3. Heat island intensity distribution”: this subsection not explained in the methodology section.

Figure 10.: “Case NoUrban”: isn’t logical to use replace a percentage of building with vegetation instead of NoUrban scenario?

Line 336: “Fig.5 and Fig.11”, do you mean Fig.10?

Data: why MODIS land use data used while higher resolution data of land use available from Sentinel 2?

Reviewer 3 Report

The paper entitled “Study on urban thermal environment in Beijing based on local climate zone method “ wants to analyze UHII and temperatures using remote sensing images. The paper is very interesting and well-structured but does not introduce any new contribution to scientific knowledge. Indeed, it doesn’t present any new findings, concepts and/or methods. The introduction reports few papers but enough to understand the background. However, I suggest citing other outstanding papers especially on UHI and UHII: https://doi.org/10.1016/j.uclim.2021.100895; https://doi.org/10.1016/j.uclim.2014.05.005; https://doi.org/10.1016/j.enbuild.2017.04.025; https://doi.org/10.3390/su13063138.  Some remarks:  all figures are not easy to read and comprehensible, consequently it is very difficult to evaluate the results reported in the paper; you shouldn’t report acronyms in the abstract; It is not clear if your model has been calibrated and validated?  Please could you comment in detail and explain well the physical parameterization in table 1? Please could you comment in detail the figure 4?   Moreover, there are many typos and errors, please you should revise carefully the paper. The conclusions are not very clear, especially the affirmation: “the fine classification of cities also affected the simulation results of temperature, but the effect was small, R increased by 0.01”.  The conclusion should be extended by adding some meaningful results and further development of the research

Round 2

Reviewer 1 Report

The authors considered the suggestions and improved the text, including additional information. The questions are answered properly, but some figures and data used still need explanations;

1-     Please add your answer on 2m Temperature to the manuscript (comment 6).

2-  In Figures 6, 7, and 8 there is an obvious difference between Case 1 - Case 2 and Case 1- Case 3 which is not discussed. 

Reviewer 3 Report

The paper deserves to be published
